# Towards Improved Field Application of Using Distributed Temperature Sensing for Soil Moisture Estimation: A Laboratory Experiment

**DOI:** 10.3390/s20010029

**Published:** 2019-12-19

**Authors:** Benjamin Apperl, Matthias Bernhardt, Karsten Schulz

**Affiliations:** Institute for Hydrology and Water Management, University of Natural Resources and Life Sciences (BOKU), 1190 Vienna, Austria; matthias.bernhardt@boku.ac.at

**Keywords:** soil water content, soil thermal properties, actively heated fiber optics, distributed temperate sensing, dual probe heat, field application

## Abstract

The “dual probe heat pulse” (DPHP) method using actively heated fiber optic (AHFO) cables combined with distributed temperate sensing (DTS) technology has been developed for monitoring thermal properties and soil water content at the field scale. Field scale application, however, requires the use of robust and thicker fiber optic cables, corroborating the assumption of an infinite thin heat source in the evaluation process. We therefore included a semi-analytical solution of the heat transport equation into the evaluation procedure in order to consider the finite thermal properties of the heating cable without a calibration procedure to estimate effective thermal properties of the soil. To test this new evaluation procedure, we conducted a laboratory experiment and tested different heating scenarios to infer soil moisture from volumetric heat capacity. Estimates were made by analyzing the shift of the temperature amplitude at the sensing cable and the characteristics of the response heating curve. The results were compared with results from the calibrated infinite line source solution and in situ water content point measurements and showed a good approximation of thermal properties for strong and short heat pulses. Volumetric water content estimates are similarly accurate to the results of the calibrated infinite line source solution. Problems arose with the cable spacing and the resettlement process after burying the cable.

## 1. Introduction

Soil thermal properties and volumetric water content are important parameters that control environmental, agricultural, and geological processes [1]. To describe heat flow, the thermal conductivity *λ* and the heat capacity *C* have to be known [2,3]. The former describes the ability of the soil to transmit heat; the latter describes its storage capacity. Both depend on soil texture, bulk density, organic matter, and soil water content [4]. The soil thermal properties can vary significantly in time and space, largely due to variations in the volumetric soil water content, which is controlled by precipitation, evapotranspiration, and the soil properties itself. These processes can vary strongly from the micro-level scale to the macro-level scale. Due to their variability and importance for heat flux forecasting, it is important to have a good knowledge of thermal properties and soil water content in any hydrological, agricultural, or water allocation model.

Current methods for estimating these variables vary in extent, support, and spacing [5]. An overview of the different measurement methods for soil water content is given in [6]. There are multiple ways of measuring the volumetric water content *ϴ*, because soil moisture controls many physical and biochemical processes. In situ point measurements with very local support are gathered by measuring dielectric properties (either time-domain-reflectometry or frequency-domain-response), electric conductivity, neutron thermalization (neutron probe), and soil thermal properties [7], or by thermogravimetric analysis. Measurements on a point scale level are widely used, and large-scale measurements can be done by remote sensing (e.g., by the SMAP satellite with a footprint of about 30 km). However, there exists a considerable gap at the intermediate scale for measuring soil water content [6]. One possibility for intermediate scale level measurements is to use ground penetrating radar [8] or electromagnetic induction [9], but they are limited to specific soil types and environmental conditions. Nevertheless, producing measurements of the soil moisture content on a field scale level are relevant for different purposes, such as agricultural management, estimation of groundwater recharge, or the general reaction of the local water availability under climate conditions.

The description of the thermal properties on a micro-scale level is done by measuring the thermal expansion of specific heat pulses. Standard sensors for measuring thermal conductivity are described in [10]. Common in situ measurements are heat flux plates or transient line source needle probes. Soil heat capacity can be measured by dual needle heat pulse probes [11,12]. However, measuring the soil thermal properties on a meso- or macro-scale level remains challenging.

One way to fill the scale level gap for measuring soil water content and thermal properties on the local scale is to build a measurement network with multiple point measurement sensors [13]. Another possibility is to use the distributed temperature sensing (DTS) technology with heated fiber optic cable [14,15,16]. The latter offers the opportunity to measure the temperature along a fiber optic cable up to several kilometers with a sampling resolution of up to 12.5 cm. In addition to fiber strength, hybrid cables contain electric conductor strength and can be heated, producing a longitudinal heat pulse. These cables are used for the single probe heat pulse probe, where one cable serves as a heater and sensor at once. Thermal conductivity can be inferred by using the analytic solution of the heat equation for a transient line heat source, which is identical to the procedure used for single needle probes [3]. The volumetric water content *ϴ* can be inferred from the soil thermal conductivity *λ*_s_ by a soil-specific transfer model predicting the soil thermal conductivity *λ*(*ϴ*) from the volumetric water content [17,18]. The dual probe heat pulse (DPHP) approach can also be adapted to fiber optic measurement by burying a heater and a sensing cable in the soil at a fixed distance. Benítez-Buelga et al. [19] showed the feasibility of the dual probe heating approach for estimating soil thermal properties using the infinite line source approach. A short heat pulse propagates from the heater cable against the sensing cable and produces a time shifted and temperature attenuated signal, depending on the thermal properties of the materials in between, and allows the measurement of *λ*_s_ and the soil heat capacity *C_s_*.

Regarding the field application of volumetric water content (VWC) measurements using DTS measurements, various problems arise: Firstly, applying the single probe approach requires a soil-specific transfer function to translate thermal conductivity to a related VWC, which hampers the applicability in heterogeneous soils. For the dual probe heat approach, no soil specific transfer function is necessary. By knowing the soil bulk density *ρd* and soil solids specific heat capacity *c_s_*, the water content can be derived from the soil heat capacity *C_s_* [20]. Secondly, a problem arising with heated fiber optic cables is the relatively large ratio of cable radii to cable spacing [19]. As a result, the cable influences itself during heat propagation, as it consists of multiple layers with varying thermal properties, and the measured heat capacity is a mixture of the heat capacity of all layers. The influence of the cable can be considered using the areal fraction of the geometry as weighted proportion to calculate the overall heat capacity of the cable. Thirdly, the standard approach for deriving thermal properties of heat pulses is the usage of the infinite line source analytic solution of the heat equation, which assumes a zero radius. However, soil DTS fiber optic cable has a relatively thick cable diameter with high crush resistance and rodent protection. Another problem might be that both single probe heat pulse and dual probe heat pulse DTS measurements suffer from poor soil-cable contact. For some sensors, the usage of porous blocks represents one possibility to eliminate this problem [21]. These problems are normally overcome by introducing an apparent spacing L_app_ to take into account the probe geometry, materials, non-zero radius, and contact resistance [22]. Gathering L_app_ requires another calibration process under known (dry or saturated) conditions, which limits the application in the field.

Towards the development of a suitable field measurement method, our study aims to improve the dual probe heat-pulse approach with fiber optic cable accounting for the influence of the cable properties on the measurement outcomes without using any calibration procedures. The main challenges to be addressed are described in [19]: the development of a model that accounts for the finite probe radii, an optimization of the heating strategies, and an improved temporal interpretation of the temperature signal. This paper addresses these questions focusing on implementing a semi-analytic solution of the heat equation, which accounts for the finite probe radius [23] of robust fiber optic cables. By having a model, which can reproduce the sensor signal adequately, the thermal response signal at the signal cable can be interpreted more precisely. It should be possible to obtain thermal properties through a regressive approach of the simulated and measured data. Additionally, the thermal impact of the cable can be quantified by comparing the results of the semi-analytic solution, which accounts for finite probe radii for the results of the line source solution with an infinite radius. To compare the results of the model to measured data, a laboratory experiment under controlled environmental conditions was conducted, testing the DPHP approach with fiber optics and validating it with independent measurements of soil moisture.

The experiments allowed us to answer an additional follow-up question: is usage of the DPHP approach with an environmentally resistant cable feasible for measuring thermal properties and soil moisture? Therefore, in a first step, the temperature signals at the sensing cables are analyzed for different heating scenarios. The best heating strategy should give a strong response signal with maximum temperature amplitude at the sensing cable, but avoiding a forced water movement. After choosing the best heating scenario, the applicability of the semi-analytic solution of the heat equation, considering the finite cable radii, is tested on its capability to simulate the influence of the cable properties properly. Therefore, the volumetric water content is calculated with this method and also the solution of the infinite line source solution of the heat equation. Afterwards they are compared with independent frequency-domain-response (FDR) volumetric water content measurements.

## 2. Materials and Methods

### 2.1. Experimental Design

A laboratory experiment was conducted using a 5 m × 0.3 m × 0.3 m perforated wooden box (Figure 1). The bottom of the box was continuously perforated with a woven filter medium attached on the upper bottom side. Above a 3 cm layer of gravel was dumped to prevent impounding of the soil layer in order to guarantee free drainage. Then, homogeneous layers of loam (as defined in Soil Science Society of America 2008) [24] with 52% sand, 34% silt, and 14% clay were topped in successive steps involving a gentle deposition by a compactor until a final layer height of 15 cm was reached. The texture of the sand was determined by particle size analysis (combined sieve and hydrometer analysis) in the laboratory. Additionally, four soil samples were taken to determine the bulk density of the soil.

During the filling process, two different fiber optic cables were positioned in the box. One centrally located hybrid fiber optic cable for heating and sensing was positioned in the middle of the box, surrounded by four sensing cable sections in a 90° angle above, to the left, to the right, and under the heating cable at a fixed distance of 18 mm soil spacing to the heating cable and a center distance L of 25 mm. Both cables were selected to withstand tough conditions in real field applications. The hybrid cable was a loose tube stainless steel cable (Silixa Ltd., Hertfordshire, UK) with a diameter of 9.1 mm consisting of two hard elastomeric tight buffered multimode fiber strength members, and two AWG 18 stranded copper wires embedded in aramid yarn strength members in a core locked flame retardant tactical polyurethane jacket (see Table 1). The copper wires were used as a heater and were coiled around a central strength member. The sensing cable, also a loose tube cable, had a diameter of 4.6 mm (BRUsens, Brugg Cable, Brugg, Switzerland) with two gel filled stranded metal loose tubes including a bend multimode fiber and four copper loam wires embedded with aramid strength members in a polyurethane outer sheath. The cable components were compactly embedded in the coating such that the air portion was negligible. The spacing of the fibers within the box was fixed by wooden spacers containing notches as large as the cable radii. To prevent longitudinal shifting or compression, the cables were fixed with clamps at the outer borders of the box [19]. During the filling process, the spacing was checked again just before burying. The relative positioning of the cables was recorded and the corresponding cable running meter of cross section profiles were ascertained every 0.5 m. The four sensor cable sections were installed with one cable looping the exit of a section with the start of the next section. The loops were longer than 1 m to avoid overlapping of the measurements of different sections. One remaining end of the sensing cable was spliced with the heating cable to obtain one single fiber optic path with both ends connected in a double ended configuration [25] to the DTS fiber optic device. Both the sensing cable and the heating cable passed through three calibration baths (two cold and one warm bath), two before entering the box, and one between the box and splice, respectively.

A Silixa Ulitma DTS fiber optic device (Silixa Ltd., Hertfordshire, UK) was used with a spatial resolution of 0.5 m and a temporal resolution of 10 s in a double ended configuration. For every cable section in the soil, ten values were obtained, but three values were excluded at the margins of the box, where the measurements might be not exclusively from the soil section. This resulted in seven measurements for the heating cable and 28 measurement values at the sensing cables. The measurement uncertainty was determined by the noise standard deviation measured in the reference ice bath sections. The temperature in the ice bath sections was recorded with an RBR SoloT high precision temperature logger with an accuracy of ±0.002 K.

In addition to the cables, eight calibrated FDR soil moisture point sensors (5TE ECH2O probes, Decagon Devices, Washington, USA) were buried during the filling process between the sensing cable and the heating cable. The positioning of the sensors was aligned with the positioning of the DTS cable to be able to allocate the FDR sensor measurements to a measured DTS section. Any influence of the sensors on the DTS measurements was assumed to be negligible, as the sensor dimensions were small in comparison to the measurement support of the fiber optics cable. The measured water content values were used as independent validation measurements.

The heating of the cable was obtained short circuiting the copper wires with a controlled current. Therefore, a high precision laboratory power supply multimeter (EA-PS 5000, EA Elektrik, Viersen, Germany) with a dynamic power control and a standard sensing input with direct connection to the load was used in order to compensate voltage drops along wire connectors.

After the soil layer was filled with the soil material, the box was flooded until free drainage conditions at the bottom were reached and the FDR sensors could measure no further rise of the volumetric water content. Saturated conditions could not be measured at all sensors. The maximum measured soil moisture content was 0.37 m^3^m^−3^.

Through evaporation processes, the soil water content decreased continuously. DTS and FDR measurements were conducted during this period. A range of measured soil water content from 0.37 to 0.06 m^3^m^−3^ was covered. The measurements started in September 2017 and lasted until the middle of November 2017 with a total of 14 measurement days. Despite the ambition to eliminate any sources of uncertainty in the experimental design, uncertainties remained. First, there were uncertainties in the positioning of the cable, which might have changed during the filling and shrinking process of the soil. The shrinking process also provoked changes of the soil bulk density. Therefore, bulk density measurements were repeated after the soil was dried up completely again. Calculations were done twice, first using both bulk densities independently, and then comparing the results. Another source of uncertainty was the changing ambient air temperature, which influenced soil temperature heat fluxes but represented real conditions for potential real time applications. As the sources of uncertainties were known and also appear in the field, they must be considered in the interpretation of the results.

### 2.2. Determination of Soil Properties

#### 2.2.1. Thermal Properties

In contrast to single probe active heated fiber optics (AHFO) with one combined heating and sensing cable [14] allowing only the determination of the thermal conductivity *λ*, dual probe heated fiber optics (DPHFO) allows the determination of thermal conductivity as well as the soil volumetric heat capacity *C* and the thermal diffusivity *κ* [19]. Uncertainties arise from variations of the experiment geometry, especially the distance between the heating and sensing cable [26], variations in the soil parameters, uncertainties in the measurements, and insufficient consideration of the finite probe radii. An analytic solution of the heat equation for the radial conduction of a heat pulse from a line source with an infinite small radius was defined [3] as
(1)$$T\left(L,t\right)=\frac{q}{4\pi \kappa C}\left\{Ei\left[\frac{L^{2}}{4\kappa (t-t_{0}}\right]-Ei\left(\frac{L^{2}}{4\kappa t}\right)\right\}$$
where *T* is the temperature at a certain time *t* and distance *L*, *t*_0_ is the heating time, *q* is the dissipated power per unit length (W/m), *κ* is the thermal diffusivity (m^2^/s), and *Ei* is the exponential integral function. However, this solution assumes a zero radius of the heater, which is practically impossible and normally considered through an apparent spacing L_app_, calibrated under known condition. To avoid this calibration procedure, we used the reformulated heat equation for an infinite line source with finite probe radii, finite heat capacity, and infinite thermal conductivity [22,23] to calculate the thermal properties. Contact resistance between cable and soil might influence the outcome but is not considered in this calculation. The analytic solution in the Laplace transform domain can be written as
(2)$$\Gamma_{2}\left(p\right)=v_{f}\left(p,r_{1},b_{1}\right)v_{f}\left(p,r_{2},b_{2}\right)\frac{\Gamma_{1}\left(p\right)K_{0}\left(mL\right)}{2\pi \lambda }$$
where Γ_2_ is the temperature at the sensor, *p* is the transform variable of the Laplace transform, *v_f_* is the transfer function, Γ_1_ is the heating function, *K*_0_ is the modified Bessel function of the second kind of order 0, *L* is the distance between the two centers of the cable, *λ* the thermal conductivity (W m^−1^ K^−1^), and
(3)$$m=\sqrt{\frac{p}{\kappa }}.$$

The transfer function *v_f_* is formulated as
(4)$$v_{f}\left(p,r_{n},b_{n}\right)={\left\{mr_{n}\left[K_{1}\left(mr_{n}\right)+\left(mr_{n}b_{n}/2\right)K_{0}\left(mr_{n}\right)\right]\right\}}^{-1}$$
where *K*_1_ is the modified Bessel function of the second kind of order 1, *r_n_* is the cable radius of the heating cable (*n* = 1) and the sense cable (*n* = 2), and *b_n_* is the ratio of the volumetric heat capacity of the cable *C_n_* divided by the volumetric heat capacity of the soil *C*. The heating function Γ of an instantaneous heating impulse of certain duration *t*_0_ is in the Laplace domain defined as
(5)$$\Gamma_{\left(p\right)}=\{\begin{array}{c} q\left[1-\exp\left(-pt_{0}\right)\right]K_{0}\left(\mathrm{mL}\right);\;\;\;\;\;\;\;\;0<t<t_{0}\\ 0\;\;\;\;\;\;\;\;\;\;\;\;\;\;\;\;;\;\;\;\;\;\;\;\;\;\;\;t\;>t_{0} \end{array}$$
where *q* is the power released per unit length cable (Wm^−1^). The analytic solution of the heat equation in the Laplace domain was evaluated using MATLAB (version 8.5.0). The inversion procedure from the Laplace domain to the time domain was done by using the Stehfest algorithm [27] with 16 coefficients, as recommended in [23]. 

The heat capacity of the cable *C_n_* was calculated by splitting the cable into its components and assigning them specific heat capacities (Table 1). In order to get the volumetric heat capacities, the subcomponents were weighed with a precision balance, and its volume was calculated from the cable geometry. The heat capacity of the cable is defined as the weighted sum of the volumetric fractions and the volumetric heat capacities of the cable components [19].

A set of seven different heating scenarios Γ_1_ is defined to test the influence of different heating scenarios on the outcomes. They differ in heating time *t*_0_ and power *q* and its range aligned to heating strategy experiences from former experiments [14,15,28]. Table 2 shows the different heating scenarios used in this experiment.

The heat impulse provokes a radial distribution of the heat with a temporal shift and an amplitude attenuation, described by the radial solution of the heat Equation (1) or (2) and generates a temperature curve Γ_2_ as a function of the of the distance and time and the soil thermal parameters *C* and *λ*. To avoid time consuming optimization procedure of *C* and *λ*, a look up table of resulting temperature functions Γ_2_(*t*) at the sensor cables were calculated within a range of potential values of *C_s_* and *λ*. For *C_s_* a range of 1 × 10^6^–5 × 10^6^ Jm^−3^K^−1^ with 10^5^ Jm^−3^K^−1^ steps was considered. Combined with *λ* ranging from 0 to 1.5 Wm^−1^K^−1^ with 0.05 Wm^−1^K^−1^ steps a total set A of 1271 temperature functions Γ_2_ at distance L result for every heating scenario Γ_1_ was used. These temperature functions were compared to the measured temperature signal from the DTS device. 

#### 2.2.2. Measured vs. Modeled Data

A classic approach of dual probe measurements is to extract the temperature amplitude *T_maxM_* at a certain time *t_maxM_* after heating has started at the sensor cable [2,19]. These values were compared to the simulated values (*T_maxS_*, *t_maxS_*) of the calculated response functions Γ_2_ in two steps. According to the law of conservation of energy and the static geometry of the system, *T_max_* only varies by variations of *C_s_*. In a first step,
(6)$$B=A\left\{\min\left|\left[T_{maxM}-T_{maxS}\left(C\right)\right]\right|\right\}$$

In a second step, *λ* was extracted in the same way from the look up table finding the best approximation, where
(7)$$D=B\left\{\min\left|\left[t_{maxM}-t_{maxS}\left(C,\lambda \right)\right]\right|\right\}$$
where D is one unique solution of the total set of solutions of A. 

A disadvantage of using this amplitude value is its difficult detection when having a measured temperature signal with proper measurement noise. The measurement noise might generate a shifted maximum peak, provoked by a positive noise spike and not only by the temperature response of the heat pulse. 

To overcome these problems, first a signal processing procedure of the raw data was done to eliminate the largest peaks. Then a correlation–regression approach of the measured data to the modeled response function Γ_2_ was conducted. The modeled values of the heating functions Γ_2_ were compared to the measured values by a best fit function using the root mean squared error. Different evaluation times t_eval_ were tested considering only the heating phase at the sensing cable, parts of the heating phase, or the heating and cooling phase for the regression approach (Figure 2). 

#### 2.2.3. Volumetric Water Content

Once the thermal properties were calculated, the volumetric water content could be inferred. The volumetric heat capacity can be formulated as the sum of the individual heat capacities of the individual constituents [20]. As the organic matter of the soil used in this experiment was infinitely small, it could be ignored. If organic matter is relevant, it has to be considered as a proper constituent. Ignoring also the heat capacity of air, which is negligible in comparison to the other constituents, the heat capacity can be expressed using the following equation [17,19]: (8)$$C=\rho_{b}c_{s}+\rho_{w}c_{w}\theta$$
where *ρd* is the soil bulk density in (kg m^−3^), *c_s_* is the soil solid specific heat capacity (J kg^−1^ K^−1^), *ρ_w_* is the density of water (kg m^−3^), *c_w_* is the specific heat capacity of water (J kg^−1^ K^−1^), and *ϴ* the volumetric water content (m^3^ m^−3^). Knowing the soil bulk density, the soil volumetric heat capacity, and the soils’ specific heat capacity, the volumetric water content can be determined by reformulating Equation (8):(9)$$\theta =\frac{C-\rho_{b}c_{s}}{\rho_{w}c_{w}}$$

The resulting volumetric water content can be compared to the corresponding values of the FDR sensors. The FDR sensors were calibrated separately with a gravimetrical soil sample analysis in the laboratory using the compacted soil. The obtained thermal properties were evaluated against the range of reasonable literature values for the corresponding soil texture [28]. Finally the semi-analytic radial solution with finite probe radii (Equation (2)) was compared to the standard heat pulse solution (Equation (1)) of [3] for a line heat source with infinite radius and a calibrated apparent cable spacing L_app_ [19]. The determination of the volumetric water content strongly depends on the accurate measurement of the heat capacity of the soil. Choosing an infinite or finite line source solution as well as using the amplitude values *T_max_* and *t_max_* or a regression approach for the determination of *C* might be decisive in the quality of the results of *ϴ*.

## 3. Results

### 3.1. Signal Interpretation

The first temperature measurements were taken under dry soil conditions (*ϴ* = 0.06). Afterwards, the soil was moistened homogeneously until saturation was registered by all FDR point sensors. The measured DTS temperature signal showed noise due to uncertainty in the measurement, which is a common problem in signal processing [29]. It was quantified with 0.04 K in the reference ice baths. The measured temperature amplitude *T_maxM_* and its corresponding time after starting the measurement *t_max_* was influenced by this uncertainty, leading to a small mismatch of the real temperature peak by overlapping of the temperature with the signal noise (Figure 2). To reduce the influence of measurement errors, noise reduction methods can be applied to the raw temperature signal. Simple smoothing with a simple moving average was considered inappropriate and would cause a flattening of the heating curve and the peak *T_max_*, as well as a possible shift of *t_max_*. Therefore, a Savitzky–Golay (SG) smoothing filter was applied to eliminate positive outliers without provoking an excessive smoothing of the temperature signal. The SG filter was specified with a polynomial function of order 3 with a frame length of seven measurements. The usage of the Savitzky–Golay filter showed satisfactory elimination of signal noise. In Figure 2, an example of the effect on the peak detection is shown, comparing the response function Γ_2_ of a 40 Watt, 150 s heat impulse without raw signal treatment (blue) and the filtered signal (red) with the modeled response function. The maximum value of the raw measurement (blue dot) was influenced by a positive outlier, increasing *T_max_* and shifting it. The smoothed Savitzky–Golay filtered signal (red) coincided better with the reference amplitude of the modeled solution (yellow dot) optimized for the heating phase.

The heating phase is well represented by the model. The cooling phase shows differences of the measured values compared to the expected modeled values with faster cooling than expected. This phenomenon was observed for different soil water contents and for rising as well as for decreasing background temperatures. A possible explanation is that heat is conducted longitudinally from the sensing cable, as this cable contains copper strength members with high conductivity, resulting in a faster temperature decrease than expected only by radial conduction. To avoid influences on the data interpretation the correlation–regression approach focused on fitting the heating phase of the temperature function Γ_2_.

Besides noise reduction models, the choice of appropriate heating scenarios is important. The influence of the measurement errors on the determination of the thermal properties decreased with an increasing ratio of the measurement error and the temperature amplitude. In Figure 3, a plot of temperature responses of different heating scenarios (Table 2) with the same volumetric water content *ϴ* is shown. Low power dissipation of short duration showed insufficient temperature response compared to the measurement uncertainty of 0.04 K. In addition, a shorter heat pulse was considered to be preferable, because it produced a slightly sharper peak at the maximum (azure vs. navy-blue) and was preferable to avoid forced water movement. Consequently, the following analysis results refer to heating scenario 7 with 40 Watt power dissipation for 150 s and a total energy dissipation of 6000 Joules.

### 3.2. Volumetric Water Content

The determination of the volumetric water content requires the measurement of the soil bulk density, the soil specific heat capacity of the solid soil phase *c_s_*, and the calculated heat capacity of the soil, and is calculated as outlined in Equation (9). The measured soil bulk density ranged from 1.45 Mg m^−3^ under dry condition before wetting to 1.6 Mg m^−3^ of the compacted, parched, dry soil. For *c_s_*, a value of 801 J kg^−1^ K^−1^ was estimated from literature values based on the results of the soil texture analysis. The volumetric heat capacity *C* was calculated as explained in Section 2.2.1. The outcomes of the volumetric water content with different bulk density and different calculation of the thermal properties were compared to the FDR sensor measurements and are shown in Figure 4. As the validation measurements refer to the volumetric water content, these results are presented first. In Section 3.3 the underlying soil properties of the best-fit solution for the FDR measurements are presented.

In Figure 4 the results of the calculated volumetric water content via DTS measurements using the semi-analytic line source solution with finite probe radii against the measured FDR measurements are shown. The usage of the semi-analytic line source solution reproduced accurate values of *ϴ*_DTS_ without calibration, independently of using the *T_maxM_* and *t_maxM_* approach (Figure 4a) or using a regression approach (Figure 4b). Figure 4a additionally shows the sensitivity of using different bulk densities, resulting in a shift of the volumetric water content. The red points show *ϴ*_DTS_ with an initially measured bulk density of 1.45 Mg m^−3^ and the blue points for a bulk density of 1.6 Mg m^−3^. Independently of *ρd*, both results had a high correlation coefficient, r = 0.84 (Table 3), but overestimated the water content. Differences were shown in the mean bias, which improved when a higher bulk density *ρd* was used. However, at higher water content, a drop of the calculated *ϴ*_DTS_ was recognized with better results from *ρd* = 1.45 Mg m^−3^. This seems logical, because of the experimental design, as the dry soil before saturation was more loosely bedded than after drying, provoked by a shrinking process of the soil during drying up. This effect was also be observed by the appearance of shrinkage cracks at a water content *ϴ* = 0.25 m^3^m^−3^ and the effect was also verified by the bulk density measurements before saturation (*ρd* = 1.45 kg dm^−3^) and after drying up (*ρd* = 1.6 kg dm^−3^). Therefore, it is recommended to use the initial bulk density for high water content and the bulk density after drying up for lower water content.

Figure 4b shows the results of the regression approach using *ρd* = 1.6 Mg m^−3^ with a considered t_eval_ of 600 s and 1000 s. *t_max_* fluctuated between 400 and 600 s. Thus, the first t_eval_ mainly included the heating phase and the second one the heating and cooling phases. Time spans between 400 and 1500 s were tested, and the best results were obtained for 600 s.

The semi-analytic line source ability to reproduce the temperature increase correctly was much better than reproducing the cooling phase, as explained in Section 3.1. Consequently, the correlation of *ϴ* with r = 0.88 and a mean bias of 0.04 was also better when considering only the heating phase of the measured data than when using the heating and cooling phases for the regression approach (r = 0.8 MB = 0.04 °C).

The regression approach with the finite line source solution over the heating phase gave the overall best results for the estimation of the volumetric water content. Using values of the consolidated bulk density gave better results for this soil type.

### 3.3. Thermal Properties

The thermal properties were calculated as explained in Section 2.2.1. We illustrated the results of the thermal properties for the methodology, which gave best volumetric water content estimates. Thermal properties were calculated using Equation (2) and the regression approach with *ρd* = 1.6 and t_eval_ = 600 s. In Figure 5, the means of the volumetric heat capacity *C* and the thermal conductivity *λ* at a given volumetric water content with their respective uncertainties are shown. The coefficient of variations ranged from 3% to 17%. The results were consistent with literature values for the same soil types at given water content [30,31]. 

However, attention has to be drawn to the drop of *C* and *λ* at high water content and the fluctuation of the resulting thermal properties. The former result is explained by the shrinking processes of the soil at the beginning of the removal of the moisture. The shrinking process increased the bulk density and, consequently also *C* and *λ*, superimposing the decrease of *C* and *λ* provoked by drying up of the soil to a water content *ϴ* of approximately 0.33 m^3^m^−3^. The latter might be explained by measurement uncertainties themselves. The support of the DTS measurement represented a 0.5 m path of the cable, although the FDR measurement supported only a single point. Additionally, the values represented the mean at a given water content, rising from different sensors, differing slightly in contact resistance, spacing, and accuracy. The FDR medium calibration accuracy was determined by ±0.02 m^3^m^−3^. Given this value, it was possible to reconfigure the results considering the accuracy of the FDR sensors (Figure 6). *C* and *λ* showed a strong correlation. Uncertainties showed slight increases at higher water contents. This was caused by the lower sensitivity of the temperature curve compared to changes at the upper range of *C* (Figure 7). 

### 3.4. Comparison to Infinite Line Source Solution

The infinite line source solution (Equation (1)) requires a calibration of the apparent spacing of the cables under well-known conditions. Otherwise, the water content differs significantly from the validation values (Figure 8). 

The calculation of *ϴ* based on Equation (1) with calibration of the apparent spacing produced feasible values (Figure 8) was almost as good as with the finite line source solution (see Table 3). Nevertheless, the validity of the data strongly depends on an accurate calibration. Whether this calibration is done at saturation or with dry soil might lead to differences in the outcome. Due to the thermal influence of the resistant cables, results without calibration result in meaningless values of VWC (Figure 8 blue).

## 4. Discussion and Conclusions

A laboratory experiment with an environmentally resistant fiber optic cable was conducted to test the practical use of the dual probe heat pulse approach with correction for the probe diameter to infer the thermal properties and the soil moisture without performing a calibration. It could be shown that it is possible to estimate the thermal properties without calibration using the semi-analytic line source solution with consideration of the finite probe radii in the case that the thermal behavior of the cable is characterized properly. By inferring *ϴ* from the volumetric heat capacity, this methodology presents an opportunity where only the soil bulk density *ρd* and the soil solids specific heat capacity *c_s_* is required a priori to get an estimate of the volumetric water content. 

The results of *ϴ* with the semi-analytic solution with finite probe radii were similar to the infinite line source solution with calibration of an apparent spacing and coincided with the measurements made with the FDR sensors. The derivation of *ϴ* from the measured values did not differ significantly when using the *T_max_* and *t_max_* regression approach. For the regression approach, it is important to include the heating phase and the peak in the time span. It is not recommended to include the cooling phase, where longitudinal thermal conduction might influence the measurements. This effect, however, is likely to be irrelevant at the field-scale, where the fiber optic cable can be implemented further outside the region of interest. Filtering of the noisy data is recommended, especially when using *T_max_* and *t_max_* for estimating *ϴ*. Regarding the heating impulse itself, a short but strong impulse is preferable. This produced a sharper temperature curve, which made the detection of *T_max_* and *t_max_* easier. The upper limit of the applied power must be chosen carefully to avoid a forced heating convection process. The estimation of *ϴ* was better at lower water contents, which arose from the higher sensibility of the temperature response curve at lower water contents. 

For field application, it is necessary to use an environmentally resistant cable. It could be demonstrated that beside the soil texture, water content, and bulk density, the cables themselves have a great influence on the temperature propagation. Consequently, it is important to consider their effects in the calculations. The infinite line source solution does not account for this influence independently if using the single probe or dual probe method. In previous studies, the effect on the outcome was reduced by choosing cables with small diameters, but hereby reducing the suitability in harsh environments. Nevertheless, when using the infinite line source solution for dual probe measurements, a calibration of the apparent radius under known soil moisture conditions and for single probe measurements, the soil specific calibration of the transfer function is needed. The usage of the semi-analytic solution with finite probe radii in combination with DTS cables overcomes these problems and makes it possible to measure the thermal properties and the water content with environmental resistant cables without a calibration procedure. Hence, this procedure represents a step towards outdoor application under real conditions. The method allows one to measure water content on multiple points along several kilometers, and although the accuracy of single point measurements is not reached, it could contribute to identify patterns of soil moisture and thermal properties effectively at the field scale.

Problems may arise with variation in cable spacing, which was attempted to be fixed with the spacers. One possibility for improvement would be the use of an aerial fiber optic cable with a fixed distance between the fiber and a heatable steel cable. Another problem could be the resettlement of the soil after burying the cable. From the experiment it can be seen that the moistening of the soil provokes shrinking and an increase of the bulk density. Consequently, for real application it is recommended that the disturbed soil layer after installing the cable is given time to restore its structure through moistening and merging processes. Another issue, which is not yet considered, is the thermal resistivity between cables and the soil, provoked by air filled cracks, which might be a reason for the remaining mean bias of *ϴ*. The optimization of the heating pulses and the improvement of the cable spacing are of interest for further studies to improve the measurement of the soil thermal properties and the soil moisture with the DPHFO technique.

## Figures and Tables

**Figure 1 sensors-20-00029-f001:**
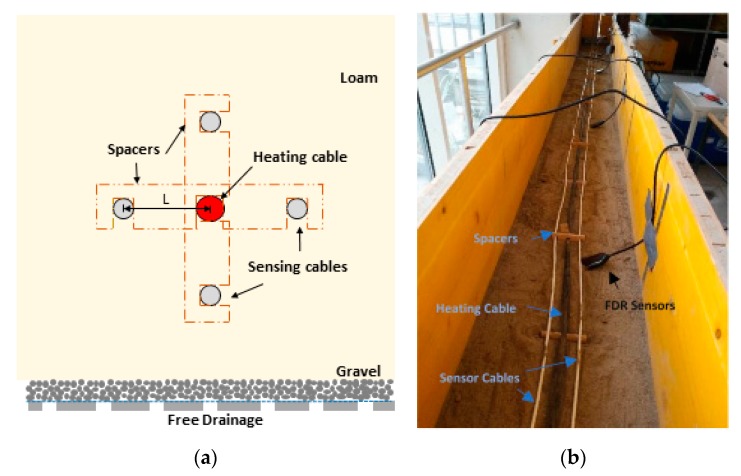
(**a**) Schematic cross section profile of cable positioning; red: heating cable, grey: sensing cables; cables are separated by wooden spacers, free drainage at the bottom to avoid impounding; (**b**) photo of the 5 m × 0.3 m × 0.3 m perforated wooden box while burying the cables. Heating cable in black, two horizontal distanced sensor cables in white. On the right the frequency-domain-response (FDR) validation sensors are buried.

**Figure 2 sensors-20-00029-f002:**
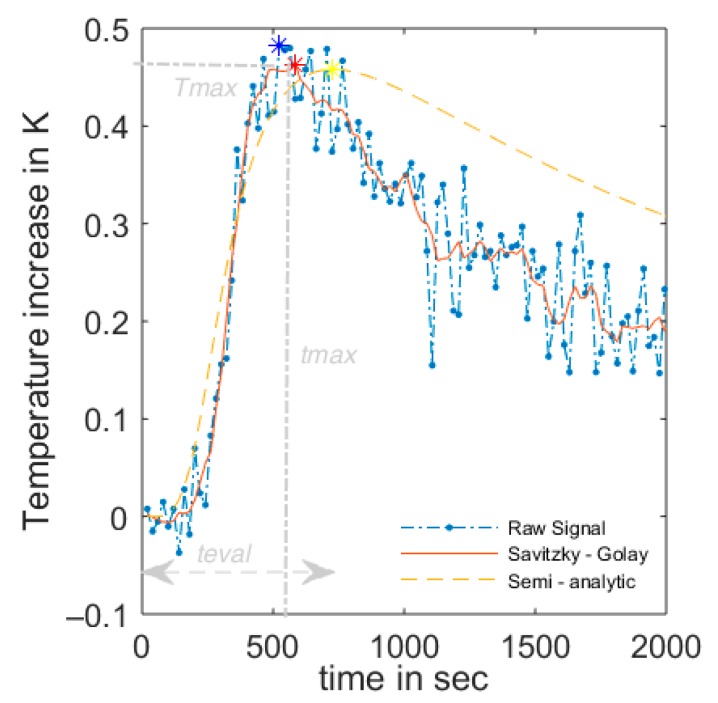
Raw signal at the sensing cable from the distributed temperate sensing (DTS) measurements with a 40 Watt heat pulse of 150 s duration in blue vs. filtered raw signals with a Savitzky–Golay Filter (3rd polynomial, seven frames) in red vs. optimum least RMSE semi-analytic solution in yellow. Points show the resulting maximum, respectively.

**Figure 3 sensors-20-00029-f003:**
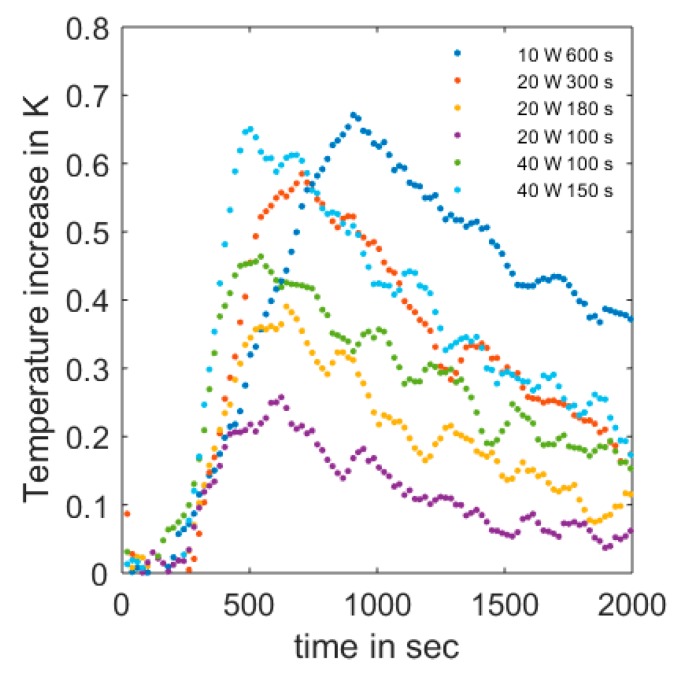
Temperature signal response at the sensing cable for different heating scenarios.

**Figure 4 sensors-20-00029-f004:**
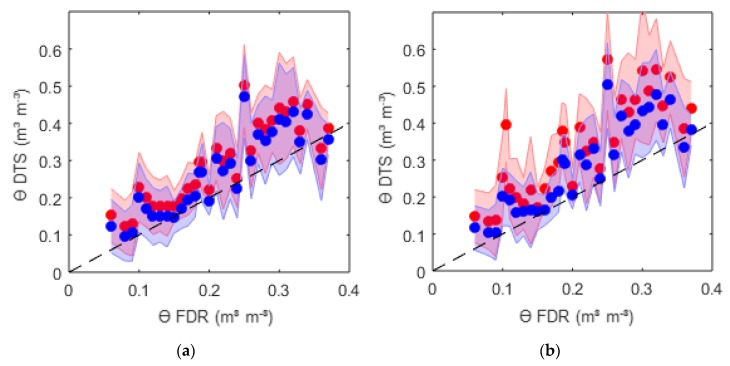
Comparison of calculated volumetric water content *ϴ* from DTS (shaded = 2 × standard deviation) measurements resulting from the finite semi-analytic solution with measurements from FDR sensor measurements for heating scenario 40 Watt and 150 s: (**a**) results extracting *T_max_* and *t_max_* for a bulk density of 1.45 Mg m^−3^ (red) and 1.6 Mg m^−3^ (blue) and (**b**) results for regression curve for t_eval_ = 1000 s (red) and t_eval_ = 600 s (blue).

**Figure 5 sensors-20-00029-f005:**
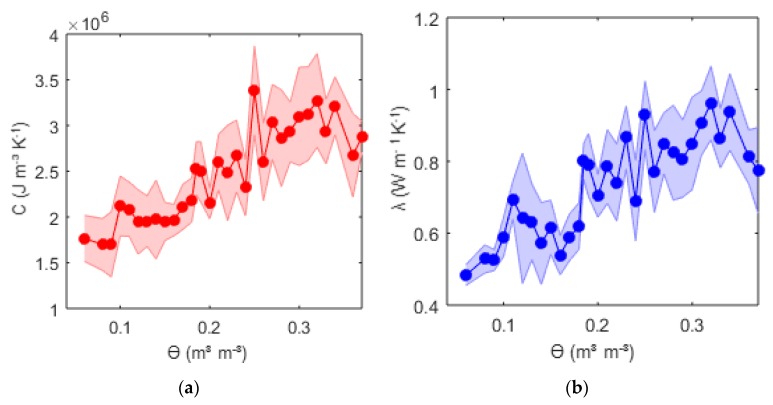
Volumetric heat capacity *C* (**a**) and thermal conductivity *λ* (**b**) vs. volumetric water content *ϴ* from the FDR sensors for a 40 Watt heat pulse of 150 s extracted from a correlation–regression approach (shaded = 2 × standard deviation of measurement uncertainties) for a 600 s time span.

**Figure 6 sensors-20-00029-f006:**
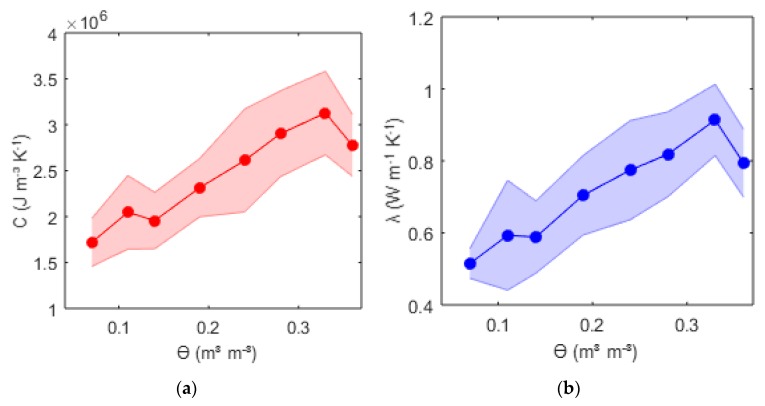
Volumetric heat capacity *C* (**a**) and thermal conductivity *λ* (**b**) vs. volumetric water content *ϴ* from the FDR sensors for a 40 Watt heat pulse of 150 s extracted from a correlation–regression approach for a 600 s time span, averaged over the medium-specific calibration accuracy of the FDR sensors of ±0.02 m^3^m^−3^.

**Figure 7 sensors-20-00029-f007:**
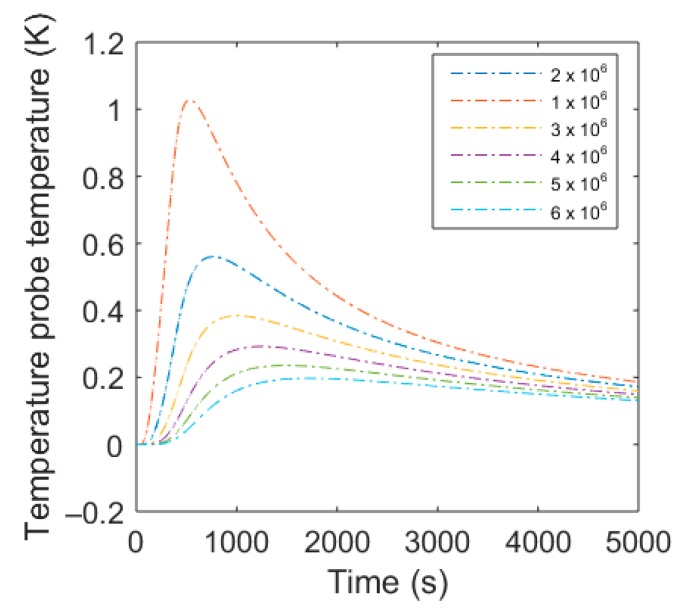
Calculated response signal with the semi-analytic solution for a 40 Watt heat pulse of 150 s for different volumetric heat capacities (1 × 10^6^ to 6 × 10^6^ Jm^−3^K).

**Figure 8 sensors-20-00029-f008:**
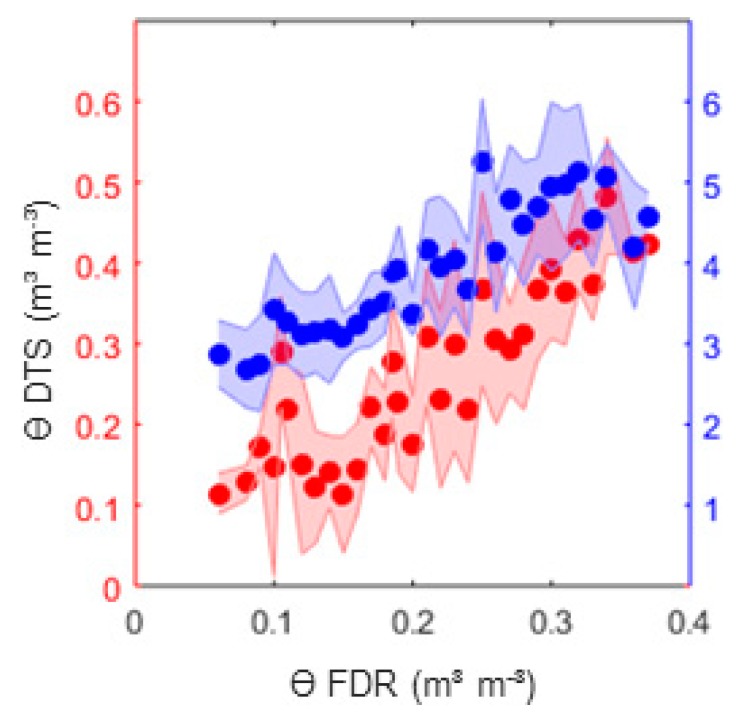
Comparison of calculated volumetric water content *ϴ* from DTS (shaded = 2 × standard deviation) measurements resulting from infinite line source solution without calibration (blue) and with calibration of the apparent spacing L_app_ (red).

**Table 1 sensors-20-00029-t001:** Thermal properties of the cable components.

Component	Material	Specific Heat Capacity c (J kg^−1^ K^−1^)	Volumetric Heat Capacity (J m^−3^ K^−1^)
**HEATER CABLE**			
Coating Heater cable	PUR ^1^	1760	2.3 × 10^6^
Aramid	Aramid	1200	2.1 × 10^2^
Copper Coating	PE ^2^	674	3.6 × 10^6^
**SENSE CABLE**			
Coating Sense cable	PUR	1760	2.3 × 10^6^
Aramid	Aramid	1200	2.1 × 10^2^
Fiber coating	Metal	510	3.9 × 10^6^

^1^ Polyurethane, ^2^ Polyethylene.

**Table 2 sensors-20-00029-t002:** Heating scenarios Γ_1._

Scenario ID	Power *q* (W m^−1^)	Heating Time *t*_0_ (s)	Total Dissipated Energy (J)
1	10	300	3000
2	10	600	6000
3	20	300	6000
4	20	150	3000
5	20	100	2000
6	40	100	4000
7	40	150	6000

**Table 3 sensors-20-00029-t003:** Correlation coefficient and mean bias of selected representative measurements of *ϴ*_DTS_ and *ϴ*_FDR._

	Finite Radius Line Source Solution				Infinite
	*T_max_ t_max_*		*Regression*		*r_app*
	*ρd* = 1.45	*ρd* = 1.6	t_eval_ = 600 s	t_eval_ = 1000 s	
*CorrCoeff*	0.85	0.85	0.88	0.8	0.82
*MB*	0.08	0.06	0.04	0.13	0.04

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
