# Peer review of "Towards Improved Field Application of Using Distributed Temperature Sensing for Soil Moisture Estimation: A Laboratory Experiment"

_sensors, 2019, doi:10.3390/s20010029_

Round 1
Reviewer 1 Report
In my opinion the paper is very well oraganized and presented and absolutely worth for publication even if it is totally technical.
Just a few comments:
Line 125: Please correct defiend--> defined
Line 493: "Multiples point". Please correct.
Thank you.
Author Response
Point1: Line 125: Please correct defiend--> defined
DONE
Point 2: Line 493: "Multiples point". Please correct.
DONE
Reviewer 2 Report
The paper presents a “dual probe heat pulse” (DPHP) method using actively heated fiber optic (AHFO) 12 cables combined with distributed temperate sensing (DTS) technology to measure thermal properties and soil water content at the field scale. A semi-analytical solution of the heat transport equation into the evaluation procedure in order to consider the finite thermal properties of the heating cable. There are some comments and questions as follows:
(1) Line 114 to 119 should be improved or rewritten. It should explain the major content of the present study.
(2) the dimension value in Fig. 1(a) is not proportional to the figure. It should be revised.
(3) the dimension and label should be added in Fig. 1(b) for better understanding.
(4) “A Silixa Ulitma DTS fibre optics device (Silixa Ltd., Hertfordshire, UK) was used with a spatial 162 resolution of 0.5 m and a temporal resolution of 10s” are you sure there is a temporal resolution?
(5) line 168, “Additionally to” should be “In addition to….”
(6) the measurement errors used in the proposed approach should be verified with other methods and discussed.
(7) The English must be improved. There are lots of grammar mistakes.
(8) The format of references needs to be revised for consistency.
Reviewer 3 Report
The manuscript describes a fibre optics sensor used to measured soil volumetric water content.
A few suggestions of small revisions that shoud be implemented:
Line 63 - ... measurement network with multiple point measurement sensors [13]. Another possibility is to use the Distributed Temperature Sensing technology (DTS) with heated fiber optics cable [14,15,16].
The Silixa Ulitma DTS is a quite expensive equiment, and perhaps other low-cost techniques of fibre optics distributed temperature sensing coud be cited, as:
F. W. D. Pfrimer, M. Koyama, A. Dante, E. C. Ferreira and J. A. S. Dias, "A Closed-Loop Interrogation Technique for Multi-Point Temperature Measurement Using Fiber Bragg Gratings," in Journal of Lightwave Technology, vol. 32, no. 5, pp. 971-977, 2014. doi: 10.1109/JLT.2013.2295536
Line 68 - "Identically to the procedure of single needle probes thermal conductivity can be inferred by using the analytic solution of the heat equation for a transient line heat source [3]. The fiber optics cable is used as a heater and as a sensor."
It is possible to understand what the authors want to say, but please rewrite to let it clear.
Line 83 - "Using the dual probe heat pulse approach is advantageous in comparison to the single probe approach since no soil-specific transfer function is required for calculating the soil water content. By knowing the soil bulk density ρd and soil solids specific heat capacity cs, the water content can be derived from the soil heat capacity Cs".
Please remember that soil space variability exists, and assuming that the soil is homogeneous in a large property is not reasonable...Also, to measured the above mentioned parameters, many laboratory experiments are required.
If space variability is present (as it is likely to be), frankly, there is not a big difference in calibrating a SHHP (within a known soil) and obtaining the soil parameters in laboratory.
Thus, it would be interesting to simply remove the statement "Using the dual probe heat pulse approach is advantageous in comparison to the single probe approach since no soil-specific transfer function is required for calculating the soil water content."
It is also important to mention in the manuscript that both the SHPP and the DHPP sufffer from the poor soil contact problem! Thus, it is worth to cite a couple of MDPI papers (one in Sensors) that use porous blocks to measure the soil water content, eliminating the soil contact problem:
1 - A Self-Powered and Autonomous Fringing Field Capacitive Sensor Integrated into a Micro Sprinkler Spinner to Measure Soil Water Content, Sensors 2017, 17, 575; doi:10.3390/s17030575
2 - Autonomous Soil Water Content Sensors Based on
Bipolar Transistors Encapsulated in Porous Ceramic Blocks, Appl. Sci. 2019, xx, 5; doi:10.3390/appxx010005.
Please note that the porous block tecnique can be employed with fibre optics sensors. The authors could make a new research project using distributed FBGs sensors on a single FO, encapsulated the FBGs in gypsum blocks. :)
In the field, the FBGs encapsulated in gypsum bocks would be deployed in the parcels of soi of a plantation, where the soil space variability requires sensors.
Line 151 - To prevent longitudinal shifting or compression the cables were fixed with clamps at the outer borders of the box [19],
Please add a comma after the word "compression".
Line 168 - Additionally to the cables eight calibrated FDR soil moisture point...
Please add a comma after the word "cables".
Line 202 - ...dual probe heated fiber optics (DPHFO) allow the determination...
...dual probe heated fiber optics (DPHFO) allows the determination
Line 290 - Ignoring the organic matter and the heat capacity of air the heat capacity can be expressed as [17,19]:
(a) Pleaase add a comma after the word "air";
(b) Is it a realistic assumption to ignore the organic content in the soil composition? Please comment in the manuscript.
1 - Characterization of a New Heat Dissipation Matric Potential Sensor, Luzius Matile *, Roman Berger, Daniel Wächter and Rolf Krebs.
Sensors 2013, 13, 1137-1145; doi:10.3390/s130101137
2 - Calibration and Temperature Correction of Heat Dissipation Matric Potential Sensors
A. L. Flint,* G. S. Campbell, K. M. Ellett, and C. Calissendorff
Soil Sci. Soc. Am. J. 66:1439–1445 (2002).
3 - Phene, C.J., G.J. Hoffman, and S.L. Rawlins. 1971. Measuring soil
matric potential in situ by sensing heat dissipation within a porous body: Theory and sensor construction.
Soil Sci. Soc. Am Proc.35:27–33.
4 - Reece, C.F. Evaluation of a line heat dissipation sensor for measuring soil matric potential.
Soil Sci. Soc. Am. J. 1996, 60, 1022–1028.
These references, together with those I have mentioned before, should make it even more clear that in the SHPP and DHPP a poor soil contact is a problem, and offer foundation to the use of porous blocks.
Round 2
Reviewer 2 Report
The authors addressed all the issues proposed by the reviewer and made good revision which is qualiftify to the journal.